# Morphology and surface chemistry engineering toward pH-universal catalysts for hydrogen evolution at high current density

Yuting Luo[1], Lei Tang[1], Usman Khan[1], Qiangmin Yu[1], Hui-Ming Cheng [1,2], Xiaolong Zou[1] & Bilu Liu[1]

Large-scale implementation of electrochemical hydrogen production requires several fundamental issues to be solved, including understanding the mechanism and developing inexpensive electrocatalysts that work well at high current densities. Here we address these challenges by exploring the roles of morphology and surface chemistry, and develop inexpensive and efficient electrocatalysts for hydrogen evolution. Three model electrocatalysts are flat platinum foil, molybdenum disulfide microspheres, and molybdenum disulfide microspheres modified by molybdenum carbide nanoparticles. The last catalyst is highly active for hydrogen evolution independent of pH, with low overpotentials of 227 mV in acidic medium and 220 mV in alkaline medium at a high current density of 1000 mA cm$^{-2}$, because of enhanced transfer of mass (reactants and hydrogen bubbles) and fast reaction kinetics due to surface oxygen groups formed on molybdenum carbide during hydrogen evolution. Our work may guide rational design of electrocatalysts that work well at high current densities.

[1] Shenzhen Geim Graphene Center (SGC), Tsinghua-Berkeley Shenzhen Institute (TBSI), Tsinghua University, Shenzhen 518055, PR China. [2] Shenyang National Laboratory for Materials Science, Institute of Metal Research, Chinese Academy of Sciences, Shenyang 110016, PR China. Correspondence and requests for materials should be addressed to X.Z. (email: xlzou@sz.tsinghua.edu.cn) or to B.L. (email: bilu.liu@sz.tsinghua.edu.cn)

The extensive use of fossil fuels has led to environmental pollution, and the need for sustainable energy storage is becoming more urgent. Electrochemical water splitting promises the production of clean hydrogen fuel from water, especially when the reaction is driven by electricity generated by wind, solar, or other renewable energy resources. However, water splitting is hindered by slow kinetics, resulting in the need for highly efficient and durable electrocatalysts. The hydrogen evolution reaction (HER) is usually initiated by the formation of adsorbed hydrogen intermediates (the Volmer step), followed by either a recombination step (the Tafel step) or an electrochemical desorption step (the Heyrovsky step) in acidic protonic media. Because of its fast kinetics in these steps[1] and good electrical conductivity, Pt has been recognized as the most efficient electrocatalyst for this reaction. Unfortunately, there are several problems preventing the widespread use of Pt-based electrocatalysts, including limited reserves and high cost. Researchers have devoted a great deal of efforts to finding Pt-free electrocatalysts, including Ru[2], transition metal disulfides[3–6], metal carbides[7,8], and metal phosphides[9,10]. For example, Mahmood et al. have reported an efficient electrocatalyst made of two-dimensional (2D) carbon decorated with Ru nanoparticles, which exhibits an activity for HER that is comparable to that of the commercial Pt/C catalyst with Tafel slopes of ~30 and ~38 mV dec$^{-1}$ in acidic and alkaline media, respectively[2]. Although the cost of Ru is only 4% of Pt, its reserve is even lower than that of Pt, making its large-scale use questionable. Therefore, earth-abundant catalysts such as $MoS_2$ have attracted intense research interest due to their low cost and high availability. Staszak-Jirkovský et al. have reported an HER electrocatalyst composed of $CoMoS_x$ chalcogels, which shows decent stability and catalytic performance with an overpotential of ~200 mV at 5 mA cm$^{-2}$ in an acidic electrolyte (pH ~1)[6]. Despite the discovery of relatively cheap electrocatalysts for HER, they are currently not viable for water splitting because they exhibit either higher overpotentials or poorer stability than Pt-based catalysts[2,6].

For practical industrial uses, the performance of electrocatalysts at large current densities is critical. For example, the current densities widely used in alkaline electrolyzers range from 200 to 500 mA cm$^{-2}$, and can reach 1000 mA cm$^{-2}$ in some cases[11]. For proton exchange membrane electrolyzers, the current densities are in the range of 1000–2000 mA cm$^{-2}$. Unfortunately, these nonprecious catalysts, and even Pt/C, operate well only at low current densities (e.g., 10 mA cm$^{-2}$), having fairly large overpotentials at high current densities[11–13]. In industry, Raney Ni is the currently-used electrocatalyst for alkaline HER and operates well at high current densities such as 500 mA cm$^{-2}$. Although it has several advantages like low cost, large surface area, and good stability, it has overpotentials of ~300–500 mV at 500 mA cm$^{-2}$ and Tafel slopes of ~90–120 dec$^{-1}$ even in concentrated 30 wt% KOH solutions[13–16]. Therefore, developing electrocatalysts that perform well at high current densities is critical for large-scale use. To this end, Chen et al. have recently reported that α-MoB$_2$ has decent catalytic activity for HER at very high hydrogen coverage, showing an overpotential less than 400 mV at 1000 mA cm$^{-2}$ in an acid medium[11]. We note that these catalysts either have large overpotentials or are only suitable for a specific pH for HER at high current densities. Developing catalysts that work well in a wide pH range is important not only for an understanding of the different HER mechanisms in acidic and alkaline media, but also for use in different pH conditions based on specific needs. This is therefore another important issue to be addressed, especially when considering the slow water dissociation kinetics in an alkaline medium for most catalysts such as Pt and $MoS_2$, which results in poorer catalytic activity in alkaline than in acidic media[17–20]. Many efforts have been devoted to

improve the HER performance of electrocatalysts in alkaline media[3,4,19,21,22]. Subbaraman et al. have studied the HER activity of Pt in an alkaline medium by decorating Pt surfaces with Ni $(OH)_2$ nanoclusters, and the resulting material shows an overpotential half that of pure Pt at 10 mA cm$^{-2}$ [17]. Later, other metal hydroxides such as $Co(OH)_2$ have also been shown to work as water dissociation promoters in alkaline media[18,22–25]. However, species that can work as kinetic promoters for water dissociation are few and are mainly limited to metal hydroxides and oxyhydroxides. Overall, the challenges in designing catalysts that work well over a range of pH values at high current densities stem from the fact that HER involves electron transfer and redistribution at liquid–solid–gas interfaces, which becomes complicated at large current densities and in different pH conditions[26]. Specifically, features of a catalyst may affect electron transfer rate, the amount and exposure of active sites, accessibility of catalytic surfaces to reactants, bonding strength with hydrogen, and water dissociation kinetics, and thus would influence their HER performance at high current densities.

Here, we address these challenges by developing electrocatalysts with an optimized morphology and surface chemistry. Three model electrocatalysts with different morphologies or/and surface chemistry are used, i.e., a flat Pt foil, $MoS_2$ microspheres made of $MoS_2$ nanosheets, and $MoS_2$ microspheres decorated by $Mo_2C$ nanoparticles ($MoS_2/Mo_2C$). Based on these studies, an efficient catalyst for HER at high current density over a range of pH values is synthesized. Microspheres are composed of radially aligned $MoS_2$ nanosheets that are decorated by $Mo_2C$ nanoparticles at their edges (denoted $MoS_2/Mo_2C$, Fig. 1a). This catalyst has many advantages. First, the aligned $MoS_2$ nanosheets have many exposed active sites that benefit in-plane electron transfer. Second, the spherical morphology has roughness at both the micro- and nano-scales, and this is necessary for access of reactants and release of hydrogen bubbles[27,28]. Third, the $Mo_2C$ nanoclusters change the surface chemistry of the $MoS_2$ catalysts. As a result, the catalyst has low overpotentials of 227 mV in acidic medium and 220 mV in alkaline medium at a high current density of 1000 mA cm$^{-2}$, small Tafel slopes of 53 mV dec$^{-1}$ (in acidic medium) and 44 mV dec$^{-1}$ (in alkaline medium), and good durability during a 24 h test in both media. Experimental and theoretical investigations show that $Mo_2C$ modified by surface oxygen groups formed during the HER not only promotes the interfacial mass transfer of reactants and hydrogen gas bubbles on $MoS_2$, but also speeds up the water dissociation and hydrogen absorption kinetics, resulting in decent HER performance at high current densities.

## Results

**Synthesis and characterization of molybdenum disulfide on molybdenum carbide.** The $MoS_2/Mo_2C$ catalyst was synthesized by a two-step method. First, microspheres of radially aligned $MoS_2$ nanosheets were grown on Ti foils by a hydrothermal method at 180 °C for 24 h. They were then loaded into a chemical vapor deposition (CVD) furnace and heated to 750 °C for the reaction with $CH_4$ to prepare $MoS_2/Mo_2C$ (see details in the "Methods" section). Scanning electron microscopy (SEM) images show that the $MoS_2/Mo_2C$ has a rugged morphology derived from the spherical $MoS_2$ (Fig. 1b, Supplementary Fig. 1). At the microscale, there are a large number of microspheres with a narrow diameter distribution of 1.61 ± 0.39 μm distributed uniformly on the surfaces of the conducting Ti foils. The inset of Fig. 1b shows that the microspheres are composed of many aligned $MoS_2$ nanosheets. Such a structure can pump liquid-phase electrolyte onto the catalytic surface because of the strong capillary forces[29]. As a result it reduces gas–solid interface

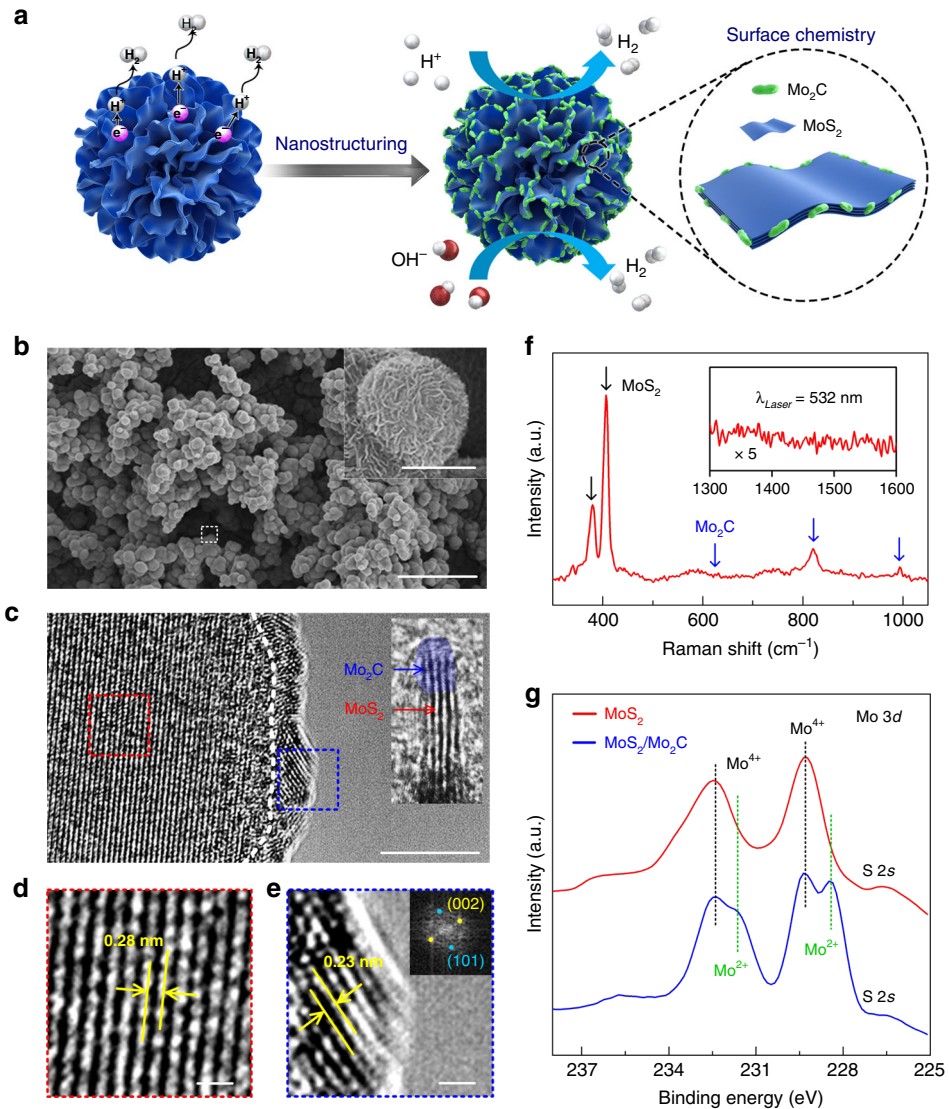

**Fig. 1** Synthesis and characterization. **a** Design schematic of the $MoS_2/Mo_2C$. **b** Scanning electron microscopy (SEM) image of the synthesized $MoS_2/Mo_2C$. The scale bar is 20 μm. The inset is an enlarged view of the dotted square and is a microsphere of $MoS_2/Mo_2C$. The scale bar is 1 μm. **c–e** High-resolution transmission electron microscopy (HRTEM) images of the $MoS_2/Mo_2C$. Insets in (**c**) and (**e**) are enlarged views and the corresponding fast Fourier transform (FFT) pattern, respectively. The images (**d**) and (**e**) are enlarged views of the squares outlined in red and blue in (**c**). The scale bars are 5 nm in (**c**) and 0.5 nm in (**d**, **e**). **f** Raman spectrum of $MoS_2/Mo_2C$. Inset shows a spectrum from 1300 to 1600 $cm^{-1}$, indicating no graphitic carbon materials were produced. **g** X-ray photoelectron spectroscopy (XPS) spectra of Mo 3$d$ of $MoS_2$ and $MoS_2/Mo_2C$. Two peaks originating from $Mo^{2+}$ appear in the $MoS_2/Mo_2C$ samples

adhesion and promotes the release of hydrogen bubbles from the catalyst surface[30], which is critical for HER involving high current densities. In addition, the aligned $MoS_2$ nanosheets have a large number of exposed catalytically active edge sites. The high-resolution transmission electron microscope (HRTEM, Fig. 1c, Supplementary Fig. 2) images show that the $Mo_2C$ nanoparticles are mainly grown on the edges of $MoS_2$ nanosheets as illustrated in Fig. 1a. This point is easy to understand because in the CVD process, the chemical conversion from $MoS_2$ to $Mo_2C$ starts at the $MoS_2$ edges where S atoms are easily attacked by hydrogen species (hydrodesulphurization), followed by carbonization in which $CH_x$ combines with the remaining $Mo$[31]. As a result, the $MoS_2$ edges are converted into $Mo_2C$ nanoparticles as shown in Fig. 1a–d. HRTEM images show typical lattice spacings of 0.28 and 0.23 nm, corresponding to the (100) plane of $MoS_2$ (Fig. 1d) and the (002) plane of β-$Mo_2C$ (Fig. 1e), respectively. The β-$Mo_2C$ structure in $MoS_2/Mo_2C$ is further confirmed by X-ray

diffraction (XRD) (Supplementary Fig. 3), and the energy dispersive spectroscopy (EDS) elemental maps show a uniform distribution of Mo, S, and C in the $MoS_2/Mo_2C$ samples (Supplementary Fig. 4).

More insight into the structure and chemical composition of the $MoS_2/Mo_2C$ samples is obtained from spectroscopic characterization. The Raman spectrum shows characteristic peaks of 2H-phase $MoS_2$ at 379 ($A_{1g}$) and 405 $cm^{-1}$ ($E_{2g}^1$) as well as β-$Mo_2C$ peaks at 660, 812, and 987 $cm^{-1}$, confirming the formation of β-$Mo_2C$ on the $MoS_2$ (Fig. 1f). Note that we do not observe any D (~1350 $cm^{-1}$) or G bands (~1590 $cm^{-1}$) associated with carbon materials, suggesting that there is no graphitic carbon formed after CVD. This is understandable because the reaction temperature is relatively low (750 °C) and no catalyst is added for the carbonization process, so that no graphitic carbon materials like graphene or carbon nanotubes are formed on the $MoS_2$. The X-ray photoelectron spectroscopy (XPS) spectra of Mo 3$d$ show

two peaks located at 229.3 and 232.4 eV from Mo $3d_{5/2}$ and Mo $3d_{3/2}$ in Mo(IV), originating from $MoS_2$. In addition, there are two peaks at 228.2 and 231.5 eV, from the Mo $3d_{5/2}$ and Mo $3d_{3/2}$ in Mo(II), suggesting the existence of $Mo_2C$ (Fig. 1g), which agrees well with the Raman, HRTEM, and XRD results. Overall, the above characterization confirms that β-$Mo_2C$ nanoparticle-modified $MoS_2$ microspheres, i.e., $MoS_2/Mo_2C$, have been synthesized by using the process illustrated in Fig. 1a.

**Multi-scale interactions between the catalyst and the hydrogen source in high current density**. Mass (liquid reactants and gas bubbles) transfers at interfaces are critical steps in the HER, especially at large current densities[32]. However, our understanding of the interactions between the source material and the catalyst at different levels is still unclear. We choose Pt foil, $MoS_2$, and $MoS_2/Mo_2C$ as model catalysts to explore the roles of morphology (micro- and nano-scales) and surface chemistry (atomic scale) on the performance of catalysts in a large current density HER. First, the HER performance of the three samples was tested in both acidic and alkaline media. All working electrodes were encapsulated (Supplementary Fig. 5) in order to ensure that the exposed surface areas were the same for the three catalysts. $MoS_2/Mo_2C$ samples were optimized by performing the CVD for different times (Supplementary Fig. 6). For a fair comparison, we tested the HER performance of Pt foil and 20 wt% Pt/C, and found that the performance of the Pt foil is comparable or better than the Pt/C (Supplementary Fig. 7 and Tables 1 and 2). Figure 2a shows the polarization curves of the three samples in a KOH (1.0 M) solution. We find that Pt foil requires a smaller overpotential to reach the same current density as $MoS_2/Mo_2C$ at current densities below 10 mA cm$^{-2}$. Note that $MoS_2/Mo_2C$ requires much smaller overpotentials to reach large current densities (e.g., 191 mV @ 500 mA cm$^{-2}$ and 220 mV @ 1000 mA cm$^{-2}$) than Pt foil (567 @ 500 mA cm$^{-2}$ and 822 mV @ 1000 mA cm$^{-2}$) and $MoS_2$ (589 @ 500 mA cm$^{-2}$ and 788 mV @ 1000 mA cm$^{-2}$), suggesting its superior performance at large current densities. As a control experiment, we sintered a $MoS_2$ sample in Ar/$H_2$ at the same conditions with the carbonization experiments, but without introducing $CH_4$ (sample denoted as $MoS_2$-H). We tested the HER performance of $MoS_2$-H and found that though catalytic performance increases a little, it is much worse than the $MoS_2/Mo_2C$, indicating $Mo_2C$ plays an important role in the good HER performance of $MoS_2/Mo_2C$ (Supplementary Fig. 8). For $MoS_2$, the HER performance is much inferior to that of a Pt foil, but as the current density increases from 0 to ~300 mA cm$^{-2}$, their difference becomes smaller and smaller, indicating the positive effect of morphology on large current density HER. These results suggest that HER performance of catalysts at different current densities are different.

To analyze the rate-determining steps for the three catalysts, Tafel plots are shown in Fig. 2b (in alkaline). Interestingly, we find that the slopes of the Pt foil and $MoS_2$ are strongly potential-dependent, but that of $MoS_2/Mo_2C$ is not (Supplementary Fig. 9)[23,33–35]. Specifically, at current densities smaller than 10 mA cm$^{-2}$, the $MoS_2/Mo_2C$ and Pt foil have Tafel slopes of 43 and 48 mV dec$^{-1}$, respectively, very close to the theoretical value of 40 mV dec$^{-1}$, where the electrochemical desorption of hydrogen (the Heyrovsky step) is the rate-limiting step[23]. As the current density increases, mass transfer plays a key role in determining the current. Therefore, we summarized the ratios of $\Delta\eta/\Delta\log|j|$ (defined as ratio, $R_{\eta/j}$, of overpotential $\eta$ to current density $j$) of three samples at different current densities to evaluate how much overpotential is needed when current increases, which could be an indicator to evaluate the performance of a catalyst at high current densities and is meaningful for

practical use (Fig. 2c). The ratio for $MoS_2/Mo_2C$ remains small (~45 mV dec$^{-1}$), but that of the Pt foil increases to more than 120 mV dec$^{-1}$ when increasing the current density. Moreover, when current density is very large (e.g., 200 mA cm$^{-2}$), the performance of Pt and $MoS_2$ is greatly affected by mass transfer at the interface, leading to much larger overpotentials needed to achieve a current density of 1000 mA cm$^{-2}$ in Pt and $MoS_2$ than in $MoS_2/Mo_2C$ (Fig. 2c). In acidic, neutral, and 6 M KOH media, the three samples show similar trends to those in the alkaline media, e.g., $MoS_2/Mo_2C$ has a small ratio (~40 mV dec$^{-1}$) in acidic media while Pt foil and $MoS_2$ have large ratios (much larger than 40 mV dec$^{-1}$) as current density increases (Supplementary Fig. 10–12). For large current density HER, $R_{\eta/j}$ should be an important indicator besides Tafel slope to evaluate the performance of electrocatalyst because it considers the influence of mass transfer on large current density HER, which is crucial for practical applications.

From the above electrochemical results, it is clear that the decoration of the $MoS_2$ by $Mo_2C$ particles has a critical influence on the interaction between catalyst and mass at interfaces, and consequently on the HER performance of the catalysts at large current densities. To quantitatively analyze the differences between the samples, we measured the contact angles (CAs) of a droplet of 1.0 M KOH on their surfaces. The CAs are 82.3°, 53.8°, and ~0° for Pt foil, $MoS_2$, and $MoS_2/Mo_2C$, respectively, indicating the best wettability of $MoS_2/Mo_2C$ by the electrolyte (Fig. 2d), which aids liquid electrolyte transfer. We recorded videos to compare the size distributions and the dynamics of the release of hydrogen bubbles on different samples, which reflects the ability to re-expose catalytic sites to the electrolyte. Clearly, as the current density increases, hydrogen bubbles firmly adhere to the Pt surface and grow to very large size (~50% are larger than 0.5 mm), covering many catalytic sites on the foil surface. In contrast, hydrogen bubbles smaller than 0.2 mm leave the surface of $MoS_2/Mo_2C$ easily, leading to the constant exposure of catalytic sites to the surrounding electrolyte (Fig. 2e, f, Supplementary Movie 1). According to the solid–liquid–gas interface theory, structures with roughness at both the micro- and nanoscale not only generate a strong capillary force to pump liquid, but also reduce interfacial adhesion to facilitate gas bubble release[29,30,36]. The structure of $MoS_2$ gives it a HER performance that is similar to flat Pt at a large current density (~400 mA cm$^{-2}$), but neither Pt nor $MoS_2$ is good enough for practical use (Fig. 2a, Supplementary Fig. 13).

To obtain high liquid and gas transfer, the interaction between the hydrogen bubbles and the catalyst at the atomic level is also important and must be considered, and this is closely related to bonding state of atoms at the catalyst surface[37]. $MoS_2$ has a relatively weak interaction with water because of its inert surface[38], while β-$Mo_2C$ shows a strong affinity to water, probably due to the formation of hydrogen bonds between its surface groups and water molecules, as in the case of MXene[39]. It is interesting to note the electrochemical surface areas of $MoS_2$ and $MoS_2/Mo_2C$ are similar (Supplementary Fig. 14), which further indicates the importance of interactions at the atomic level to mass transfer (Supplementary Movie 2). Therefore, the correct wettability of $MoS_2/Mo_2C$ to the electrolyte and hydrogen gas can produce better liquid electrolyte and gas transfer at the micro-, nano-, and atomic scales in $MoS_2/Mo_2C$ than for Pt and $MoS_2$ (Supplementary Fig. 15), resulting in better HER performance at large current densities. The interaction between the catalyst $j$ and the hydrogen source at the atomic level is important for many electrocatalysts, e.g., $MoS_2$ and graphene, and modifying them using species with a good water affinity at the atomic level may be a promising strategy to improve the performance of HER and other catalytic reactions. Although

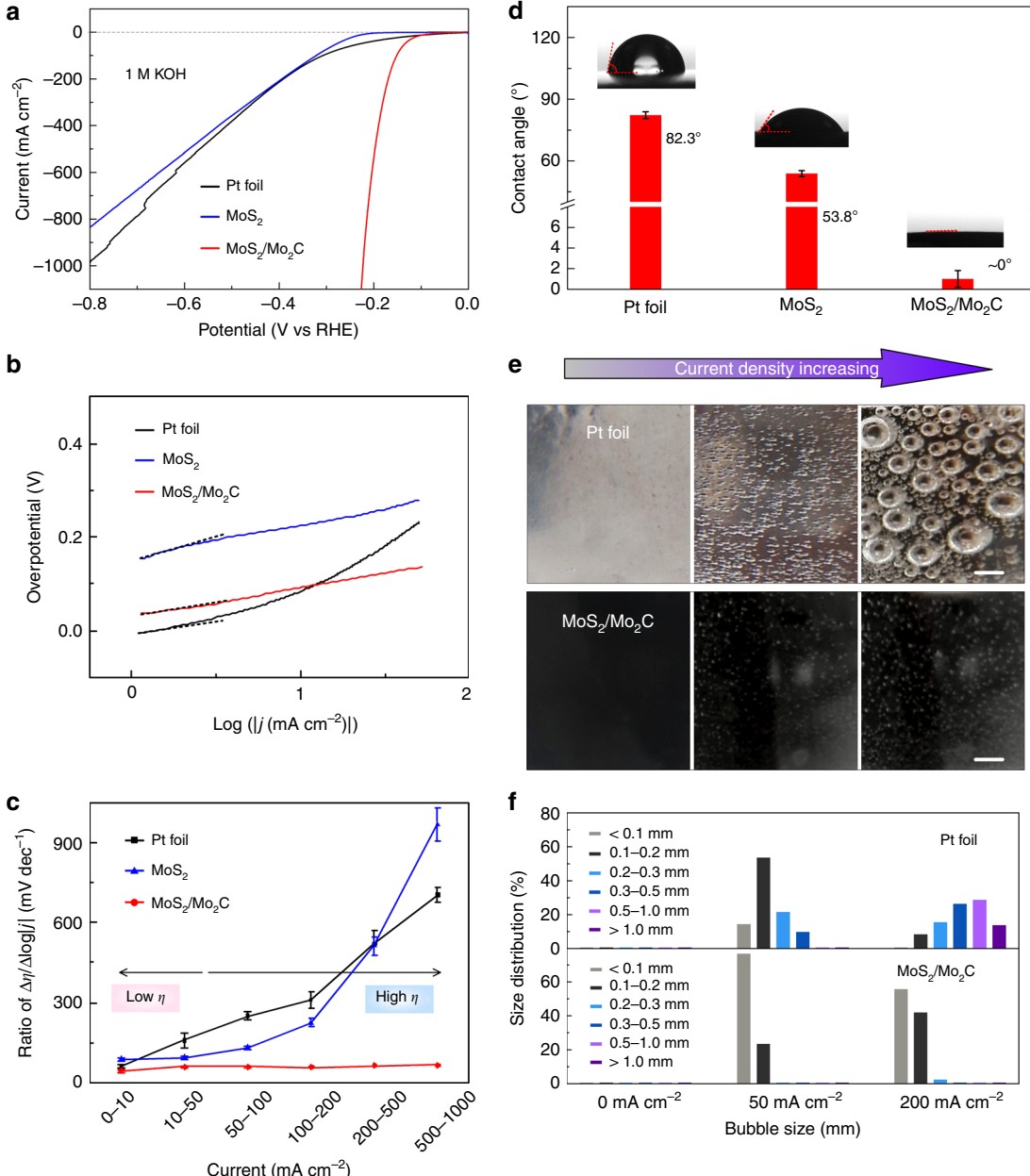

**Fig. 2** Electrocatalytic performance of different catalysts at high current densities. **a** Polarization curves and **b** Tafel curves after iR compensation for a Pt foil, $MoS_2$, and $MoS_2/Mo_2C$ in KOH (1 M) at a scan rate of 5 mV s$^{-1}$. **c** Ratios of $\Delta\eta/\Delta\log|j|$, i.e., $R_{\eta/j}$, for the three catalysts in different current density ranges, which can be used as an indicator to evaluate the performance of a catalyst at high current densities. All points were tested three times, and error bars correspond to standard deviations. Source data are provided as a Source Data file. **d** CAs of a KOH (1 M) droplet on the surfaces of the catalyst. The CAs were measured for at least three times for each sample, and error bars correspond to standard deviations. Source data are provided as a Source Data file. **e** Photos show sharp contrast during the release of $H_2$ bubbles on the Pt foil and on $MoS_2/Mo_2C$ surfaces. The scale bars are 1 mm. **f** Size distributions of $H_2$ bubbles on the surfaces of a Pt foil and $MoS_2/Mo_2C$. Source data are provided as a Source Data file

the mass transfer ability at catalytic interfaces can be improved at certain degree by passive ways like stirring or pumping electrolytes, rationally design of electrocatalysts could be more energy-saving and efficient.

**Fast water dissociation kinetics with electrocatalysts**. The overpotentials at 1000 mA cm$^{-2}$ and the Tafel slopes of three samples are given in Fig. 3a, b. It is apparent that Pt and $MoS_2$ have a higher catalytic performance in acidic than in alkaline media because slow water dissociation is the rate-determining step for Pt and $MoS_2$[17,20,24]. $MoS_2/Mo_2C$ shows decent and

comparable HER performance in both media, suggesting a great increase in catalytic performance in alkaline media after the $MoS_2$ has been modified by β-$Mo_2C$. Nyquist plots of the three samples show that $MoS_2/Mo_2C$ has better electron transfer ability than the other two samples in both media (Supplementary Fig. 16), which is in accordance with the catalytic performance of three samples. For a practical electrocatalyst, it is important to maintain good stability over long-term use, and this is very challenging especially at large current densities. The polarization curve of the $MoS_2/Mo_2C$ sample shows a negligible shift after 10,000 cycles, suggesting its excellent HER stability in both media (Fig. 3c). It also shows good HER performance after a 24 h test at a large current

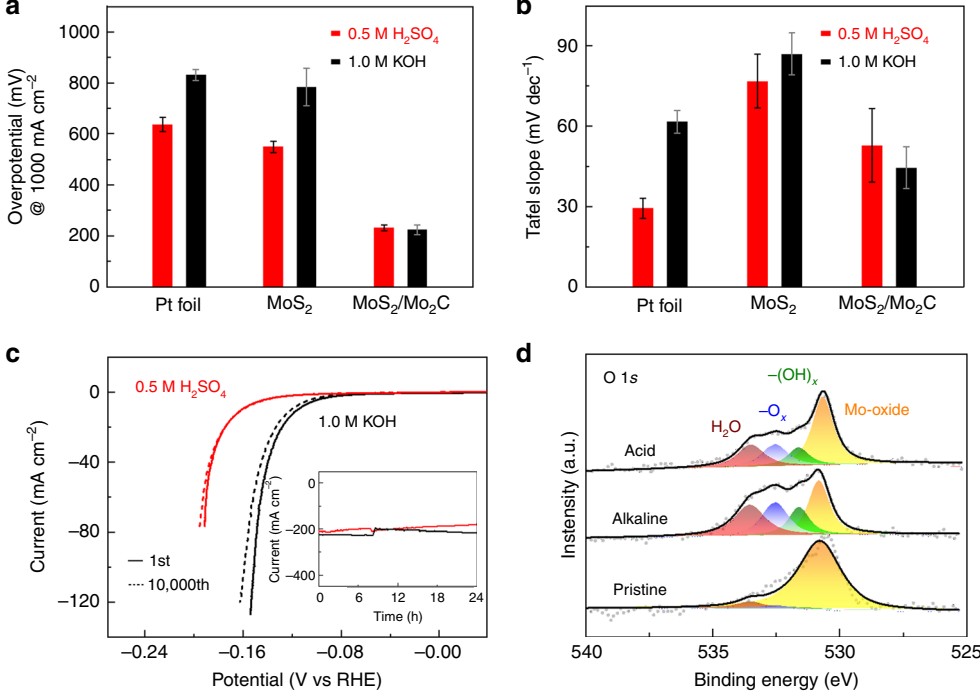

**Fig. 3** Performance and surface chemistry of catalysts. **a** Overpotentials at 1000 mA cm$^{-2}$ for a Pt foil, MoS$_2$, and MoS$_2$/Mo$_2$C in acidic and alkaline media. Each sample was measured for three times, and error bars correspond to standard deviations. Source data are provided as a Source Data file. **b** Tafel slopes (at a current density smaller than 50 mA cm$^{-2}$) for a Pt foil, MoS$_2$, and MoS$_2$/Mo$_2$C in acidic and alkaline media. Each sample was measured for three times, and error bars correspond to standard deviations. Source data are provided as a Source Data file. **c** Polarization curves of the MoS$_2$/Mo$_2$C catalyst during the initial scan and after 10,000 scans. Inset shows the chronoamperometric responses (i–t) recorded on MoS$_2$/Mo$_2$C for 24 h in both media. **d** O 1s X-ray photoelectron spectroscopy (XPS) spectra of the MoS$_2$/Mo$_2$C sample before and after 100 cycles in KOH (1.0 M) and H$_2$SO$_4$ (0.5 M) solutions

density of 200 mA cm$^{-2}$ (inset in Fig. 3c). All electrochemical results confirm that MoS$_2$/Mo$_2$C shows good catalytic performance, including a small overpotential, a small Tafel slope, and good durability, at large current densities in both acidic and alkaline media.

The MoS$_2$/Mo$_2$C sample shows comparable HER performance in both alkaline and acidic media, but the Pt foil and MoS$_2$ samples show poorer HER performance in alkaline than in acidic media, which stems from differences in their surface chemistry. To reveal the role of surface chemistry on water dissociation kinetics, we conducted O 1s high-resolution XPS of the MoS$_2$/Mo$_2$C catalyst before and after 100 cycles in acidic and alkaline media. All samples were carefully treated to ensure that they were exposed to the ambient environment for less than 1 min (see Methods, Fig. 3d, and Supplementary Fig. 17 and 18). The peaks at 530.8, 531.6, 532.5, and 533.5 eV are assigned to oxygen ions (O$_{lattice}$) in molybdenum oxides, Mo$_2$C(OH)$_x$ (–OH terminated), Mo$_2$CO$_x$ (–O terminated), and Mo$_2$C(OH)$_x$–H$_2$O (–OH terminated with strongly adsorbed water), respectively[38,39,40]. Note that after cycling in acidic or alkaline media, the surface of β-Mo$_2$C is modified by terminating –OH and –O groups. This phenomenon was also found in other materials in previous studies, such as Mo$_2$C and Ti$_3$C$_2$ in MXenes[39,41]. In contrast, neither Mo 3d nor S 2p peaks of the MoS$_2$ show any observable changes, suggesting that there is no obvious change of MoS$_2$, such as proton intercalation or surface oxygen modification during the above electrochemical process (Supplementary Fig. 19)[3]. These results suggest that Mo$_2$C particles modified by surface oxygen were formed during HER and work as a water dissociation promoter, resulting in greatly increased HER catalytic performance in alkaline media. Compared to metal hydroxide promoters that are only stable in alkaline media, electrocatalysts modified by Mo$_2$C are suitable for use in both acidic and alkaline

media. Moreover, Mo$_2$C is metallic with a similar electronic structure to Pt. Considering the metallic nature of Mo$_2$C with an electronic conductivity of 10$^6$ S m$^{-1}$ [42], much higher than those of insulating metal hydroxides (10$^{-3}$–10$^{-2}$ S m$^{-1}$)[43], it is reasonable to argue that Mo$_2$C would serve as a better promoter and assist electrochemical hydrogen evolution.

**Formation of surface oxygen and mechanism for pH-universal hydrogen evolution.** To identify the oxygen species formed on Mo$_2$C surface during HER, we used pure β-phase Mo$_2$C as a model material, and the oxygen species formed on Mo$_2$C surface were analyzed by high-resolution XPS. Figure 4a shows the relationship between the electrochemical potential, acidic or alkaline environment, and oxygen species on the surface of β-Mo$_2$C. It is evident that as the electrochemical potential increases, the main peak of surface oxygen changes from –OH terminated species to –O terminated species (Fig. 4a, b). For example, when the pH is 14, the Mo$_2$C surface is –O terminated at a potential of −0.25 V vs a reversible hydrogen electrode (RHE) and become –OH terminated at a potential of −1.5 V vs RHE. At pH = 0, the trend is similar. To further understand the relationship between electrochemical potential, pH, and oxygen species on Mo$_2$C surface, we performed density functional theory (DFT) calculations and obtained a Pourbaix diagram, which shows the most stable oxygen species on Mo$_2$C surfaces as a function of pH and electrochemical potential (Fig. 4c for the (101) surface, Supplementary Fig. 20 for the (001) surface). By comparing high-resolution XPS with the Pourbaix diagram, stable phases of the Mo$_2$C (101) surface at different potentials and pH values show good agreement with the experimental results in Fig. 4a. Accordingly, in acidic media, –OH terminated Mo$_2$C (101) surfaces play a key role in determining the catalytic performance. In

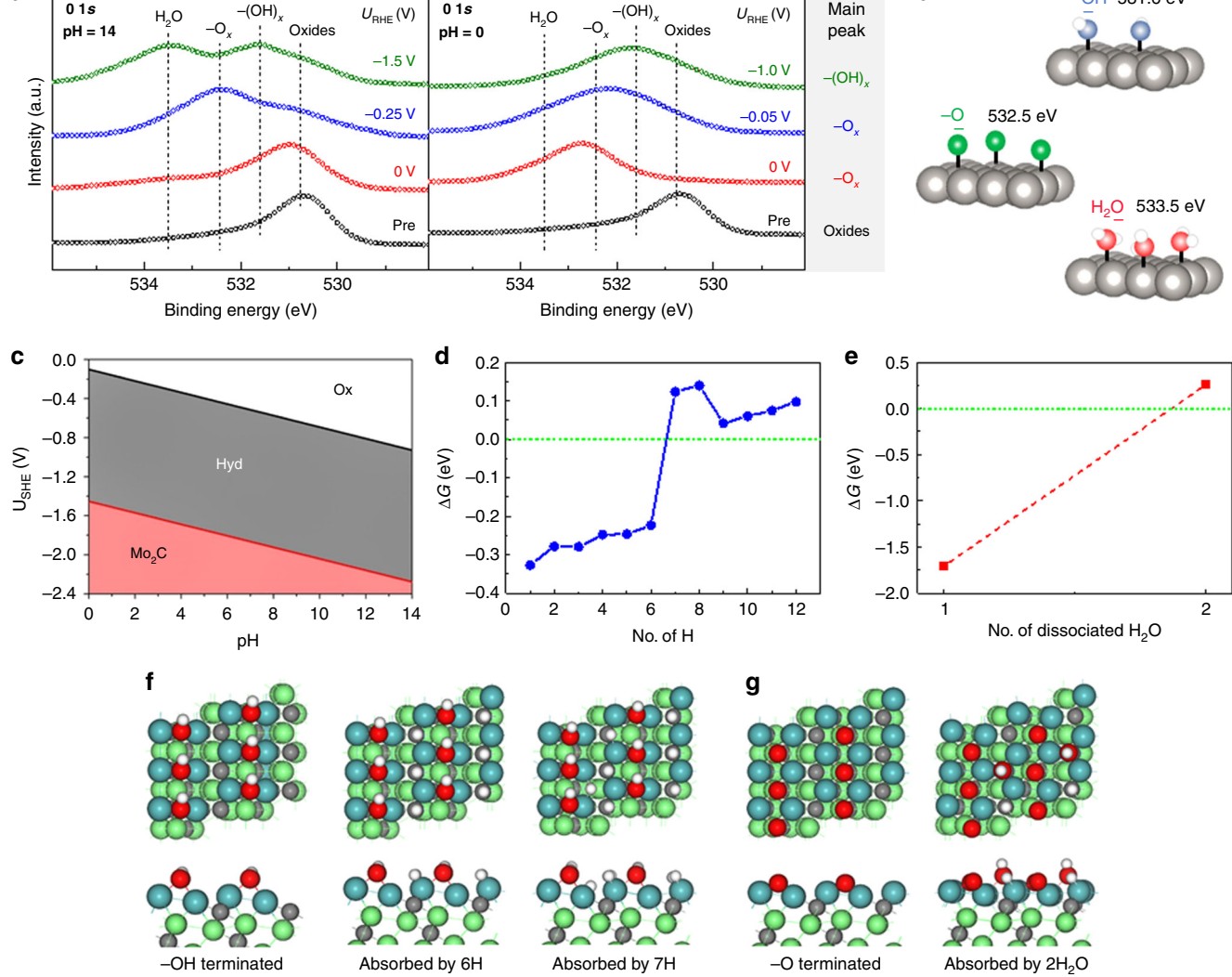

**Fig. 4** Self-optimized surface oxygen on the (101) surface of β-Mo₂C to enable fast kinetics. **a** X-ray photoelectron spectroscopy (XPS) spectra of β-Mo₂C that has undergone the hydrogen reduction reaction at different pH values and potentials. **b** Models and binding energies of surface oxygenated species, including –OH (blue, 531.6 eV), –O (green, 532.5 eV), and H₂O (red, 533.5 eV). **c** Pourbaix diagram of Mo₂C (101) at different pH values and potentials. Here, "Ox" and "Hyd" stand for –O and –OH terminated Mo₂C surfaces, while "Mo₂C" represents the free surface. **d–g** DFT calculations showing adsorption energies for **d** H on –OH terminated, and **e** disassociated H₂O on –O terminated Mo₂C (101) as a function of adsorbed species. **g**, **f** The top and side views of the corresponding optimized structures, where (dark) green, gray, red, and white spheres represent Mo, C, O, and H atoms, respectively

contrast, in alkaline media, –O terminated Mo₂C (101) surfaces play a key role. To explain the catalytic performance of these surfaces in different media, Fig. 4d shows the calculated adsorption energies for H on a –OH terminated surface as a function of the number of adsorbed H atoms (the results for H on a –O terminated surface is shown as Supplementary Fig. 21), while Fig. 4e shows the calculated adsorption energies for disassociated H₂O (OH and H) on an –O terminated surface as a function of the number of adsorbed H₂O molecules. It can be seen that a hydrogen free energy of ~0.1 eV and a small energy barrier for water dissociation (~0.25 eV) are achieved, corresponding to fast hydrogen absorption/desorption and water dissociation kinetics. The relaxed structures of H on –OH terminated, and disassociated H₂O on –O terminated Mo₂C (101) are shown in Fig. 4f, g. These theoretical results explain the improved HER performance of MoS₂/Mo₂C compared to MoS₂ in both acidic and alkaline media. These results are interesting for two reasons. First, the surface oxygen formed on β-Mo₂C during HER changes the electron distribution on β-Mo₂C surface, which

highlights the importance of the interaction between the catalyst and the electrochemical conditions on its performance. Second, the oxygen-terminated β-Mo₂C surface theoretically increases the interfacial mass transfer of the catalyst because oxygen terminates have a strong affinity to water and weak affinity to gas bubbles. It is possible that this could be further developed to design even better electrocatalysts than MoS₂/Mo₂C.

## Discussion

We have explored the roles of the morphology and surface chemistry of catalysts on HER at large current densities using three model electrocatalysts, i.e., a flat Pt foil, MoS₂ microspheres made of MoS₂ nanosheets, and the same MoS₂ microspheres decorated by Mo₂C nanoparticles (MoS₂/Mo₂C). The MoS₂/Mo₂C electrocatalyst is highly active and stable, and is a catalyst for HER at all pH values, performing especially well at a large current density of 1000 mA cm⁻². Experimental and theoretical investigations indicate that improved interfacial mass transfer

and surface oxygen formed on β-Mo$_2$C during HER are two crucial reasons for the good HER performance of MoS$_2$/Mo$_2$C. First, in addition to structures with both micro- and nanoscale roughness, surface chemistry at the atomic level is crucial for interfacial mass transfer at large current densities. This finding suggests that electrocatalysts should be modified with species with good water affinity at the atomic level. Second, the interactions between electrochemical conditions and Mo$_2$C lead to self-optimized oxygen on Mo$_2$C surface, where oxygen-terminated Mo$_2$C (101) shows fast kinetics for both hydrogen absorption/desorption and water dissociation. These findings shine new light on the effect of morphology and surface chemistry on HER performance especially at large current densities, and pave the way to design good electrocatalysts for practical HER use. Furthermore, the discovery of oxygen-terminated β-Mo$_2$C for promoting water dissociation can be used to design other HER electrocatalysts that work well in both acidic and alkaline media.

## Methods

**Synthesis of molybdenum disulfide microspheres**. We grew MoS$_2$ microspheres on a Ti foil by a hydrothermal method. First, a piece of Ti foil (purity >99.99%, 40 × 30 × 0.5 mm) was cleaned with concentrated HCl (37 wt%) for 30 min, then bath sonicated in deionized water and ethanol, each for 5 min. Second, amine molybdate ((NH$_4$)$_6$Mo$_7$O$_{24}$•4H$_2$O, 0.1766 g) and thiocarbamide (CS(NH$_2$)$_2$, 0.484 g) were added to deionized water (36 mL) and stirred to form a clear solution. Finally, the aqueous solution contains the Mo and S precursors and the Ti foil were transferred to a Teflon-lined stainless-steel autoclave (50 mL), maintained at 180 °C for 24 h to grow MoS$_2$ on the Ti foil[33]. The Ti foil with a black film on its surface was taken out of the autoclave and thoroughly rinsed with deionized water and ethanol, and then dried in vacuum at 60 °C for 6 h. We used the hydrothermal method to synthesize MoS$_2$ because a little oxygen can be introduced into MoS$_2$ during the synthesis, which would increase the HER performance of MoS$_2$[44].

**Synthesis of molybdenum disulfide on molybdenum carbide**. The Ti foil with MoS$_2$ grown on it was put into a 1.5 in. diameter horizontal quartz tube furnace that was heated to 750 °C with a mixture of Ar (100 standard cubic centimeter per minute (sccm)) and H$_2$ (30 sccm) in 30 min. CH$_4$ (10 sccm) was then introduced into the tube furnace and the CVD was performed at 750 °C for different times (0, 20, 60, or 100 min) to prepare the MoS$_2$/Mo$_2$C samples. After the reaction, the CH$_4$ was turned off and the furnace was cooled to room temperature under Ar (100 sccm) and H$_2$ (30 sccm).

**Materials characterization**. The morphology of the samples was examined by SEM (5 kV, Hitachi SU8010, Japan). TEM and HRTEM analyses were carried out at an electron acceleration voltage of 300 kV (FEI Tecnai F30, USA). Raman spectra were collected using 532 nm laser excitation with a beam size of ~1 μm (Horiba LabRAB HR800, Japan). Structural and chemical analyses of the samples were performed by powder XRD (Cu Kα radiation, λ = 0.15418 nm, Bruker D8 Advance, Germany) and high-resolution XPS (monochromatic Al Kα X-rays, Thermo Fisher ESCALAB 250Xi, England). The pass energy was 20 eV and energy step size was 0.1 eV. Oxygen species on the surface of the Mo$_2$C were analyzed by high-resolution XPS after the samples had been collected and dried inside an Ar-filled glove box. The samples were exposed to air for less than 1 min before the XPS measurements to avoid oxidation in an ambient environment. XPS of each sample that experienced HER at a particular potential and pH was performed at least twice. The CAs of droplets on the sample surfaces were recorded by a contact angle measuring device (MDTC-EQ-M07-01, Japan). The droplet volume was the same in each case.

**Electrochemical measurements**. The MoS$_2$ and MoS$_2$/Mo$_2$C catalyst loadings were measured to be about 0.3 mg cm$^{-2}$. A standard three-electrode electrolyzer with KOH (1.0 M) or H$_2$SO$_4$ solution (0.5 M) was used in all tests, with a saturated calomel electrode (SCE) and a graphite rod as the reference and counter electrodes, respectively. The scan rate was 5 mV s$^{-1}$ for linear sweep voltammetry tests and 50 mV s$^{-1}$ for long-term cyclic voltammetry tests. Before each test, the electrolyte was bubbled with Ar for 15 min to remove dissolved oxygen in the solution.

**Density functional theory calculations**. Using the Vienna Ab-initio Simulation Package[45,46], all calculations were performed using the Perdew–Burke–Ernzerhof version[47] of generalized gradient approximation for DFT. The ion–electron interaction was described by projector-augmented wave potentials[48,49]. A symmetric (2 × 3) Mo$_2$C (101) slab consisting of five layers of Mo and four layers of C, and a symmetric (3 × 3) Mo$_2$C (001) slab composed of three layers of Mo$_2$C were constructed as our models. During structural relaxation, the bottom two layers of Mo and C for the (101) slab and the bottom Mo$_2$C layer for the (001) slab were

fixed. In the aperiodic direction, a vacuum layer larger than 10 Å was selected to keep spurious interactions negligible. The force convergence for structural relaxation was set to be 0.01 eV/Å. The thermodynamic free energies $G = E_{DFT} + E_{ZPE} - TS$ were determined following the computational hydrogen electrode model[50]. Here, $E_{DFT}$ and $E_{ZPE}$ are the DFT ground state and zero-point energies. The molecular entropy was taken from the standard tables for gas-phase molecules[50], and the vibrational entropy for adsorbed species was calculated as $S = k_B \left[ \sum_i ln \left( \frac{1}{1-e^{-h\nu_i/k_BT}} \right) + \sum_i \frac{h\nu}{k_BT} \frac{1}{(e^{h\nu/k_BT}-1)} + 1 \right]$[51]. For the Pourbaix diagram, the adsorption energies of O and OH were obtained taking H$_2$O and H$_2$ as references, based on equations (6) and (7) in ref.[50].

## Data availability

All data are available from the authors upon reasonable request.

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

## Acknowledgements

The authors thank Prof. Hengqiang Ye for giving suggestions on the analyses of TEM data, and Prof. Wei Lv and Mr. Shujie Xiao for their assistance in carbonization experiments. The authors acknowledge support by the National Natural Science Foundation of China (No. 51722206), the Youth 1000-Talent Program of China, the Economic, Trade and Information Commission of Shenzhen Municipality for the "2017 Graphene Manufacturing Innovation Center Project", the Shenzhen Basic Research Project (Nos. JCYJ20170307140956657 and JCYJ20170407155608882), and the Development and Reform Commission of Shenzhen Municipality for the development of the "Low-Dimensional Materials and Devices" discipline.

## Author contributions

Y.L., L.T., X.Z., and B.L. conceived the idea. Y.L. and L.T. synthesized the catalyst and performed XRD, Raman, and SEM characterization. Y.L. performed TEM, XPS, CAs, and electrochemical tests. U.K. took part in the analysis of XRD and TEM results. Q.Y. took part in the electrochemical data discussion. X.Z. performed the theoretical calculations. B.L. and X.Z. supervised the project and directed the research. Y.L., L.T., X.Z., H.-M.C., and B.L. interpreted the results and wrote the manuscript with feedbacks from the other authors.
