## [Peer Review File · Nature Communications]

Reviewers' comments:

Reviewer #1 (Remarks to the Author):

This manuscript reports on the synthesis of MoS₂ microspheres modified with Mo₂C particles to create a composite that is active for the HER in acidic and alkaline media. In addition the material is stable when operated at high current densities up to 1000 mA cm⁻². The presence of Mo₂C was the key component to ensuring activity in both media whereby surface oxygen was postulated to promote fast electron transfer kinetics for hydrogen adsorption/desorption and water dissociation. Overall the study is carried out well and offers new insights into the HER at these types of materials. However, the following points should be considered:

A key aspect of this work is the role of surface oxygen groups on Mo₂C and how they influence electrocatalytic activity. This was undertaken at beta-phase Mo₂C to correlate with DFT calculations. However this model discounts the role that MoS₂ may also be playing and its interaction with Mo₂C? XPS characterisation of the sample should therefore be performed after the HER and compared to only Mo₂C to probe if the same effect is occurring and how that is influenced by the presence of MoS₂. In addition the S 2p XPS spectra should also be analysed before and after HER.

Recent work on MoS₂ materials (J. Am. Chem. Soc. 2013, 135, 17881–17888) has indicated that oxygen induced defects promote HER activity by increasing the conductivity of MoS₂ and also creating more active sites – could this also be playing a role in the increased activity given the presence of Mo-oxide peaks in the XPS spectrum?

Another possible factor is intercalation of protons into the structure during the HER in acidic media as reported in J. Am. Chem. Soc. 2017, 139, 16194–16200 – could this also be considered in this work?

Is the material active in neutral conditions?

On a technical note it should be noted that these samples are exposed to atmosphere for ca. 1 min which may be a significant factor in influencing the surface chemistry. This is particularly important after the HER which may activate the surface and make it more reactive to ambient oxygen. Given the critical role that surface oxygen is playing in this work, XPS characterisation needs to be performed for a material that has not been exposed to atmosphere.

The material can operate at high current densities but some context should be given as to the typical current densities used in commercial alkaline electrolyzers. In addition commercial electrolyzers generally operate at much higher KOH concentration (can be ca. 6 M) – have the authors considered testing their material under such conditions?

Reviewer #2 (Remarks to the Author):

This work studied the roles of morphology and surface chemistry in electrocatalytic hydrogen evolution and reported the development of highly efficient catalyst, MoS₂/Mo₂C, for HER at large current density and wide range of pH. On the one hand, the morphology with roughness at both the micro- and nano-scales could enhance the access of reactants and release of hydrogen bubbles. On the other hand, Mo₂C modified by surface oxygen formed during the HER not only promotes the interfacial mass transfer of reactants and hydrogen gas bubbles on MoS₂, but also speeds up the water dissociation and hydrogen absorption kinetics. This work is interesting and instructive, but there are still some

detailed problems to be rectified and revised.

1. Page 6: "Scanning electron microscopy (SEM) images show that the MoS₂/Mo₂C has a rugged morphology derived from the spherical MoS₂ with Mo₂C particles between the layers." "The high resolution transmission electron microscope (HRTEM) images show that the Mo₂C nanoparticles are mainly grown on the edges of MoS₂ nanosheets."

The description of the site of Mo₂C is confusing. And it's hard to see Mo₂C particles and impossible to see layers in SEM image. According to the TEM images and the synthesis mechanism stated in page 6, Mo₂C is more likely to be at the edge of MoS₂.

2. Figure 1g: "In addition, there are two peaks at 228.2 and 231.5 eV, from the Mo 3d_{5/2} and Mo 3d_{3/2} in Mo(II), suggesting the existence of Mo₂C."

These peak positions are very close to Mo metal (228 eV). So they should be ascribed to Mo^{δ+}, not Mo(II).

3. Why the HER performance of MoS₂/Mo₂C in alkaline solution is better than that in acidic solution?

4. What's the mechanism for the formation of –O or –OH groups on Mo₂C surface.

5. Figure 3d: In the acid curve, the separated peak for –OH is very broad, which is inappropriate.

6. Page 16: "Accordingly, –OH and –O terminated Mo₂C (101) surfaces play a key role in determining the catalytic performance in acidic and alkaline media, respectively."

According to Figure 4c, "Ox" is formed at low overpotential while "Hyd" is formed at higher overpotential, so both Ox and Hyd should be active in HER, in spite of pH.

Reviewer #3 (Remarks to the Author):

In this manuscript, the authors investigated the electrochemical HER performance of three model electrocatalysts of flat Pt, MoS₂ microspheres, MoS₂ microspheres decorated with Mo₂C nanoparticles (MoS₂/Mo₂C). They found that the MoS₂/Mo₂C is highly active for HER in universal electrolyte, with low overpotential of 227 mV in acid and 220 mV in base electrolyte to drive a large current density of 1000 mA cm⁻². By exploring the roles of morphology and surface chemistry, authors argued that the microstructure and surface oxygen formed on Mo₂C under HER working potential are two reasons for the high activity of MoS₂/Mo₂C. While the HER catalytic performance is good and the surface chemistry design is interesting, the underlying mechanism study is still superficial and questionable. So, the referee would like to suggest a major revision based on the status of the manuscript and high quality requirement for publication of Nature Communications. The detailed comments and questions are listed as following.

1: For the catalytic performance, it is unfair to compare their activity by geometric surface area, especially for mechanism study. It is suggested to compare the activity of catalysts based on electrochemical surface area.

2: It is suggested to provide LSV curves of the three electrocatalysts with and without IR correction.

3: Maybe I am wrong, but the Tafel equation can only be applied to electrochemical polarization control situation. Therefore, I suppose it is meaningless to discuss rate determining step based on Tafel slope at very high overpotential with high current density.

4: In the manuscript, the authors discussed the bubble size on the flat Pt foil and MoS₂/Mo₂C to emphasize the important role of microstructure on HER performance. Instead, I am more interested in the difference of bubble behavior between MoS₂/Mo₂C and MoS₂ microspheres because they have similar microstructure but huge difference in activity.

5: In the experimental part, I found that the MoS₂ microspheres were synthesized through hydrothermal reaction without high temperature sintering, while the MoS₂/Mo₂C microspheres were sintered in Ar/H₂ gas and CH₄ gas within different times. As we both know that the sintering of MoS₂ in H₂ gas can produce defects in MoS₂ with high HER activity, it is uncertain to attribute the improvement of HER activity of MoS₂/Mo₂C electrocatalyst barely to the decoration of Mo₂C.

6: It is interesting to find that the authors assume the decorated Mo₂C nanoparticles as active sites for HER, while the ECSA of MoS₂/Mo₂C and MoS₂ microspheres are similar and the Mo₂C nanoparticles are only decorated on the terminal of MoS₂ nanosheets. Generally, people consider the active sites are linearly related with ECSA values. As MoS₂/Mo₂C and MoS₂ microspheres have similar amount of active sites, where is the improved activity come from?

7: It is hard to understand the surface -O have higher binding energy than surface -OH in XPS, due to H⁺ has electron withdraw effect. Is it possible that the samples had been contaminated in the glovebox, as the O1s binding energy at 532.5 eV is more likely assigned as organic (-C=O-) bonds? Maybe the authors can check out with flat Pt or MoS₂/Mo₂C Pre in glove box with the same time.

8: For DFT calculation, it is recommended to calculate the H absorption energy on the surface of benchmarking Pt catalyst as well, by doing so that can verify the model and methodology for your calculation.

9: In base electrolyte, the surface of Mo₂C catalyst are terminated with -O groups. For HER activity in base media, H₂O dissociation is only half of the HER reaction coordination. It is suggested to calculate H absorption energy on -O terminated Mo₂C as well.

10. As the Mo₂C nanoparticles were barely decorated on the terminal of MoS₂ nanosheets and contributed negligible increase of ECSA (as to say related with active sites), would this be reasonable to consider Mo₂C as active sites for HER?

Response to Reviewer #1

Comment. This manuscript reports on the synthesis of MoS₂ microspheres modified with Mo₂C particles to create a composite that is active for the HER in acidic and alkaline media. In addition the material is stable when operated at high current densities up to 1000 mA cm⁻². The presence of Mo₂C was the key component to ensuring activity in both media whereby surface oxygen was postulated to promote fast electron transfer kinetics for hydrogen adsorption/desorption and water dissociation. Overall the study is carried out well and offers new insights into the HER at these types of materials. However, the following points should be considered:

Response. Thank you very much for your positive recommendations.

Comment 1. A key aspect of this work is the role of surface oxygen groups on Mo₂C and how they influence electrocatalytic activity. This was undertaken at beta-phase Mo₂C to correlate with DFT calculations. However this model discounts the role that MoS₂ may also be playing and its interaction with Mo₂C? XPS characterisation of the sample should therefore be performed after the HER and compared to only Mo₂C to probe if the same effect is occurring and how that is influenced by the presence of MoS₂. In addition the S 2p XPS spectra should also be analysed before and after HER.

Response 1. This is a very good point. We actually have already shown the XPS spectra in our original manuscript. From Figure 3d (MoS₂/Mo₂C sample) and Figure 4a (Mo₂C sample), the two samples show similar peaks in the O 1s XPS spectra after HER, showing the same effect is occurring. These results suggest that the existence of MoS₂ in MoS₂/Mo₂C does not affect the formation of surface oxygen groups on Mo₂C. In addition, according to your suggestions, we added the S 2p XPS spectra of MoS₂ samples before and after HER (Figure R1). The spectra show negligible changes of the S 2p peaks, suggesting that MoS₂ is quite stable during HER tests. Accordingly, we believe that MoS₂ does not play any important role in surface oxygen modification of Mo₂C. In the MoS₂/Mo₂C composite catalyst, the main role of MoS₂ microsphere is to promote the fast release of H₂ bubbles at large current densities.

Figure R1. S 2p XPS spectra of MoS₂ samples before and after HER tests, showing negligible changes of the peaks. This figure was added as Figure S19a in the revised Supplementary Information.

Comment 2. Recent work on MoS₂ materials (*J. Am. Chem. Soc.* **2013**, *135*, 17881–17888) has indicated that oxygen induced defects promote HER activity by increasing the conductivity of MoS₂

and also creating more active sites – could this also be playing a role in the increased activity given the presence of Mo-oxide peaks in the XPS spectrum?

Another possible factor is intercalation of protons into the structure during the HER in acidic media as reported in *J. Am. Chem. Soc.* **2017**, *139*, 16194–16200 – could this also be considered in this work?

Response 2. We have read these two papers carefully, and considered the two points you mentioned. First, the catalytic performance of MoS₂ can be improved by oxygen induced defects and it is easy to introduce oxygen into MoS₂ during hydrothermal synthesis. This is one of the reasons why we choose hydrothermal method to synthesize MoS₂. Our XPS data also shows that a little oxygen has been introduced into MoS₂, agreeing with the paper you mentioned (*JACS*, 2013). Because MoS₂ itself shows a much poorer performance than MoS₂/Mo₂C composites in our work, we believe that the major contribution to the greatly improved performance is from the decoration of Mo₂C, rather than the introduction of oxygen-related defects in MoS₂ during synthesis.

Second, regarding proton intercalation. In our samples, neither S 2p nor Mo 3d XPS spectra of MoS₂ show observable changes before and after HER cycling in acidic media (Figure R2). Therefore, we believe that proton intercalation does not happen in our samples. The MoS₂ we used is synthesized in a different way with the paper (*JACS*, 2017), which might be a reason for the different results obtained.

Changes to the revised manuscript. We have added a sentence on Page 20 in the revised manuscript. “We used the hydrothermal method to synthesize MoS₂ because a little oxygen can be introduced into MoS₂ during the synthesis, which would increase the HER performance of MoS₂.⁴⁵” On Page 15, “In contrast, neither Mo 3d nor S 2p peaks of the MoS₂ show any observable changes, suggesting that there is no obvious change of MoS₂, such as proton intercalation or surface oxygen modification during the above electrochemical process (Supplementary Fig. S19).⁴²”

Figure R2. (a) S 2p and (b) Mo 3d XPS spectra of MoS₂ samples before and after HER cycling, showing no obvious changes of the binding energies. This figure was added as Figure S19 in the revised Supplementary Information.

Comment 3. Is the material active in neutral conditions?

Response 3. We have tested the HER performance of the materials in a neutral condition (0.5 M Na₂SO₄). As shown in the Figure R3, the MoS₂/Mo₂C shows a smaller overpotential (449 mV) than Pt foil (636 mV) at 10 mA cm⁻².

Figure R3. Polarization curves with and without iR correction for a Pt foil and a MoS₂/Mo₂C sample in a neutral condition (0.5 M Na₂SO₄). The scan rate is 5 mV s⁻¹. The MoS₂/Mo₂C shows a smaller overpotential (449 mV) than Pt foil (636 mV) at 10 mA cm⁻². This figure was added as Figure S11 in the revised Supplementary Information.

Comment 4. On a technical note it should be noted that these samples are exposed to atmosphere for ca. 1 min which may be a significant factor in influencing the surface chemistry. This is particularly important after the HER which may activate the surface and make it more reactive to ambient oxygen. Given the critical role that surface oxygen is playing in this work, XPS characterisation needs to be performed for a material that has not been exposed to atmosphere.

Response 4. This is a very good point. We have followed your suggestions and conducted XPS characterization of a Mo₂C sample (right after HER) without expose to atmosphere. Briefly, a vacuum transfer stage was used to load the samples inside Ar-filled glove box. Later, the vacuum transfer stage was put inside the chamber of XPS instrument for measurements. So the sample was not exposed to atmosphere during the whole process. We found that the O1s XPS spectra of Mo₂C after the HER tests in alkaline and acidic media (Figure R4) are very similar to those shown in Figure 4a, confirming that the surface oxygen groups on Mo₂C were not related to ca. 1 min air exposure.

Figure R4. O 1s XPS spectra of β -Mo₂C samples that first underwent the HER tests and were then directly transferred to XPS chamber for measurements using a vacuum transfer stage. Note that the samples were not exposed to atmosphere during the whole process. The results show very similar

oxygen peaks to those shown in Figure 4a. This figure was added as Figure S17 in the revised Supplementary Information.

Comment 5. The material can operate at high current densities but some context should be given as to the typical current densities used in commercial alkaline electrolyzers. In addition commercial electrolyzers generally operate at much higher KOH concentration (can be ca. 6 M) – have the authors considered testing their material under such conditions?

Response 5. Thank you for this kind suggestion. We have added some context in the revised manuscript to discuss the typical current densities used in commercial alkaline electrolyzers, as you suggested. In addition, we tested the HER performance of the MoS₂/Mo₂C and Pt foil at high KOH concentration (6 M KOH, Figure R6) and added the results in the revised Supplementary Information, as you suggested.

Figure R6. Polarization curves with and without iR correction for Pt foil and MoS₂/Mo₂C in concentrated KOH (6 M) solutions. The scan rate is 5 mV s⁻¹. This figure was added as Figure S12 in the revised Supplementary Information.

Changes to the revised manuscript. On Page 3. “For practical industrial uses, the performance of electrocatalysts at large current densities is critical. For example, the current densities widely used in alkaline electrolyzers are from 200 to 500 mA cm⁻², and can reach 1000 mA cm⁻² in some cases.^{11,21} For proton exchange membrane electrolyzers, the current densities are in the range of 1000 to 2000 mA cm⁻².”

Response to Reviewer #2

Comment. This work studied the roles of morphology and surface chemistry in electrocatalytic hydrogen evolution and reported the development of highly efficient catalyst, MoS₂/Mo₂C, for HER at large current density and wide range of pH. On the one hand, the morphology with roughness at both the micro- and nano-scales could enhance the access of reactants and release of hydrogen bubbles. On the other hand, Mo₂C modified by surface oxygen formed during the HER not only promotes the interfacial mass transfer of reactants and hydrogen gas bubbles on MoS₂, but also speeds up the water dissociation and hydrogen absorption kinetics. This work is interesting and instructive, but there are still some detailed problems to be rectified and revised.

Response. Thank you very much for your positive recommendations.

Comment 1. Page 6: “Scanning electron microscopy (SEM) images show that the MoS₂/Mo₂C has a rugged morphology derived from the spherical MoS₂ with Mo₂C particles between the layers.” “The high resolution transmission electron microscope (HRTEM) images show that the Mo₂C nanoparticles are mainly grown on the edges of MoS₂ nanosheets.”

The description of the site of Mo₂C is confusing. And it’s hard to see Mo₂C particles and impossible to see layers in SEM image. According to the TEM images and the synthesis mechanism stated in page 6, Mo₂C is more likely to be at the edge of MoS₂.

Response 1. Yes, you are right. Mo₂C particles are mainly grown on the edges of MoS₂ nanosheets, as illustrated in Figure 1a. To make this point more clear, we changed the descriptions on Page 6 in the revised manuscript to be: “Scanning electron microscopy (SEM) images show that the MoS₂/Mo₂C has a rugged morphology derived from the spherical MoS₂” and “The high resolution transmission electron microscope (HRTEM, **Fig. 1c, Supplementary Fig. S2**) images show that the Mo₂C nanoparticles are mainly grown on the edges of MoS₂ nanosheets as illustrated in Figure 1a”.

Comment 2. Figure 1g: “In addition, there are two peaks at 228.2 and 231.5 eV, from the Mo 3d_{5/2} and Mo 3d_{3/2} in Mo(II), suggesting the existence of Mo₂C.” These peak positions are very close to Mo metal (228 eV). So they should be ascribed to Mo ^{II}, not Mo(II).

Response 2. We guess that the reviewer’s question is about how to distinguish Mo(II) and metallic Mo(0) from the XPS data. Actually, it is difficult to distinguish them solely by XPS because their peaks are very close, as you pointed out. Therefore, we combined multiple techniques together in our work, including XPS, Raman, HRTEM, and XRD. First, the Raman spectrum shows peaks of Mo₂C (Figure 1f), confirming the existence of Mo₂C. Second, from the HRTEM, we only found Mo₂C and MoS₂, but did not find Mo metals (Figures 1c-1e). Third, the XRD patterns show peaks originated from MoS₂ and Mo₂C, but no peaks from Mo metals (Figure S3). Even though we cannot exclude the formation of trace amount of Mo(0) from the above results, we believe that the majority of Mo are in the state of Mo(II) in Mo₂C.

Comment 3. Why the HER performance of MoS₂/Mo₂C in alkaline solution is better than that in acidic solution?

Response 3. To compare their relative performance, Figure R7 shows calculated H adsorption energies on either –O or –OH terminated Mo₂C (101) surface, which are the dominant surface under alkaline or acidic solutions, respectively. To make a fair comparison, we first benchmarked our results from PBE functional with respect to well-accepted ones for Pt (111) using RPBE functional (This is also suggested by Reviewer 3#). We found that a correction of 0.1 eV is needed for PBE calculated results. After including the correction, the results show that H adsorption energies on –O terminated Mo₂C (101) surface are closer to 0 than those on –OH terminated surface at high hydrogen coverage cases. This could be one reason that MoS₂/Mo₂C shows better performance in alkaline than in acidic solution.

Figure R7. DFT calculated adsorption energies for H on $-O$ (Ox) or $-OH$ (Hyd) terminated Mo_2C (101) surfaces. Ox-corr and Hyd-corr stand for the adsorption energies corrected by a shift of 0.1 eV for PBE functional. The results show that H adsorption energies on $-O$ terminated Mo_2C (101) surface are closer to 0 than those on $-OH$ terminated surface at high hydrogen coverage cases, suggesting higher performance of MoS_2/Mo_2C in alkaline media than in acidic media. This figure was added as Figure S21 in the revised Supplementary Information.

Comment 4. What's the mechanism for the formation of $-O$ or $-OH$ groups on Mo_2C surface?

Response 4. As can be seen from the Pourbaix diagram of Mo_2C at different pH values and potentials (Figure 4c and Figure S20), the formations of $-O$ or $-OH$ groups on Mo_2C surface are more thermodynamically favorable in these electrochemical conditions.

Comment 5. Figure 3d: In the acid curve, the separated peak for $-OH$ is very broad, which is inappropriate.

Response 5. Thank you for your kind suggestion. We have carefully refitted the curve in Figure 3d in the revised manuscript to give a better fitting (Figure R8). After refitting, the results did not show influence to the original conclusion, *i.e.*, after cycling in acidic or alkaline media, the surface of MoS_2/Mo_2C sample is modified by $-OH$ and $-O$ groups.

Figure R8. Refitting of the O 1s XPS spectra of the MoS_2/Mo_2C sample after 100 cycles in H_2SO_4 (0.5 M, acid) solutions. This figure was added as Figure 3d in the revised manuscript to replace original Figure 3d.

Comment 6. Page 16: “Accordingly, –OH and –O terminated Mo₂C (101) surfaces play a key role in determining the catalytic performance in acidic and alkaline media, respectively.”

According to Figure 4c, “Ox” is formed at low overpotential while “Hyd” is formed at higher overpotential, so both Ox and Hyd should be active in HER, in spite of pH.

Response 6. Thank you for pointing this out. Yes, you are right. According to our DFT calculations, both –OH and –O terminated Mo₂C (101) are active in HER. In addition, combining the HER testing conditions with the Pourbaix diagram, Mo₂C (101) surfaces are mainly modified by –O surface group in the acidic media, while major –OH and minor –O surface groups in the alkaline media. Thus, we conclude that –OH and –O terminated Mo₂C (101) surfaces play a key role in determining the catalytic performance in acidic and alkaline media, respectively. To avoid any ambiguity, we rewrote this sentence on Page 17 in the revised manuscript to be: “Accordingly, in acidic media, –OH terminated Mo₂C (101) surfaces play a key role in determining the catalytic performance. In contrast, in alkaline media, –O terminated Mo₂C (101) surfaces play a key role.”

Response to Reviewer #3

Comment. In this manuscript, the authors investigated the electrochemical HER performance of three model electrocatalysts of flat Pt, MoS₂ microspheres, MoS₂ microspheres decorated with Mo₂C nanoparticles (MoS₂/Mo₂C). They found that the MoS₂/Mo₂C is highly active for HER in universal electrolyte, with low overpotential of 227 mV in acid and 220 mV in base electrolyte to drive a large current density of 1000 mA cm⁻². By exploring the roles of morphology and surface chemistry, authors argued that the microstructure and surface oxygen formed on Mo₂C under HER working potential are two reasons for the high activity of MoS₂/Mo₂C. While the HER catalytic performance is good and the surface chemistry design is interesting, the underline mechanism study is still superficial and questionable. So, the referee would like to suggest a major revision based on the status of the manuscript and high quality requirement for publication of Nature Communications. The detailed comments and questions are listed as following.

Response. Thank you very much for the reviewer’s positive recommendations about the interesting chemistry design and catalytic performance of our work.

Comment 1. For the catalytic performance, it is unfair to compare their activity by geometric surface area, especially for mechanism study. It is suggested to compare the activity of catalysts based on electrochemical surface area.

Response 1. According to the reviewer’s comment, we measured the electrochemical surface area of the Pt foil. The results show that the double layer capacitance of Pt is much smaller than those of MoS₂ and MoS₂/Mo₂C (Figure R9). This result indicates that Pt foil shows better intrinsic activity per active site than MoS₂ and MoS₂/Mo₂C, in accordance with our claim in the manuscript that “Pt foil shows a smaller overpotential at 10 mA cm⁻² than MoS₂ and MoS₂/Mo₂C (on Page 9)”. We agree with the reviewer that Pt has a higher intrinsic activity. In our work, we focus on the overall catalytic performance (not solely the intrinsic activity) of the catalysts, because the overall catalytic performance is more meaningful for practical use. The overall catalytic performance of a catalyst is determined by the intrinsic activity of each active site, number of active sites, and mass transfer

behavior (especially at large current densities). To make our statement more clear and avoid any misleading, we changed “catalytic activity” to “catalytic performance” in the revised manuscript.

Figure R9. (a, c, e) CV curves at different scan rates. (b, d, f) CV and capacitive current plotted as a function of scan rate in 1.0 M KOH at scan rates of 10, 20, 40, 60, 80, 100, and 200 mV s⁻¹ for (a, b) MoS₂, (c, d) MoS₂/Mo₂C, and (e, f) Pt foil. The results show that the C_{dl} values for the MoS₂ and MoS₂/Mo₂C, which are proportional to the electrochemical surface area, are very close (37.68 and 37.98 mF cm⁻²), suggesting similar numbers of catalytically active sites in the two electrocatalysts for HER. The C_{dl} of Pt foil (0.47 mF cm⁻²) is much less than MoS₂ and MoS₂/Mo₂C, suggesting better intrinsic per-site activity of Pt than MoS₂ and MoS₂/Mo₂C. The better intrinsic activity of Pt leads to better HER activity of Pt foil at relatively low current density of 10 mA cm⁻² than the other two catalysts. This figure was shown in Figure S14 in the revised manuscript.

Comment 2. It is suggested to provide LSV curves of the three electrocatalysts with and without iR correction.

Response 2. According to your suggestions, we have provided the LSV curves of the three catalysts with and without iR correction in the revised Supplementary Information (Figure R10).

Figure R10. Polarization curves in (a) 1.0 M KOH and (b) 0.5 M H₂SO₄ for MoS₂/Mo₂C, MoS₂, and a Pt foil without (dotted lines) and with (solid lines) iR correction. This figure was added as Figure S13 in the revised Supplementary Information.

Comment 3. Maybe I am wrong, but the Tafel equation can only be applied to electrochemical polarization control situation. Therefore, I suppose it is meaningless to discuss rate determining step based on Tafel slope at very high overpotential with high current density.

Response 3. This is a very interesting point. Yes, the Tafel equation can only be used in the electrochemical polarization controlled situation. We noticed this point and that is why we used “slope”, instead of “Tafel slope”, in Figure 2c. The purpose of Figure 2c is to show how much potential is needed when increasing the current, which could be an indicator to evaluate the performance of a catalyst at large current densities and is meaningful for practical use. To avoid any ambiguity, we replotted Figure 2b (to cut the large current density regime), and changed “slope” to “Ratios of $\Delta\eta/\Delta\log|j|$ ” in Figure 2c, and re-wrote some texts in the revised manuscript. Of course, if the reviewer still thinks it is not proper, we can further revise or delete Figure 2c based on your suggestions.

Changes to the revised manuscript. On Page 10. “As the current density increases, mass transfer plays a key role in determining the current. Therefore, we summarized the ratios of $\Delta\eta/\Delta\log|j|$ of three samples at different current densities to evaluate how much overpotential is needed when current increases, which could be an indicator to evaluate the performance of a catalyst at large current densities and is meaningful for practical use (Fig. 2c). The ratio for MoS₂/Mo₂C remains small (~ 45 mV dec⁻¹), but that of the Pt foil increases to more than 120 mV dec⁻¹.” On Page 13. “(c) Ratios of $\Delta\eta/\Delta\log|j|$ for the three catalysts in different current density ranges, which can be used as an indicator to evaluate the performance of a catalyst at large current densities.”

Comment 4. In the manuscript, the authors discussed the bubble size on the flat Pt foil and MoS₂/Mo₂C to emphasize the important role of microstructure on HER performance. Instead, I am more interested in the difference of bubble behavior between MoS₂/Mo₂C and MoS₂ microspheres because they have similar microstructure but huge difference in activity.

Response 4. This is a nice suggestion. We have performed experiments and captured the bubble behaviors of MoS₂ and MoS₂/Mo₂C using in-situ optical microscopy (Movie S2). We compared the bubble sizes of Pt foil, MoS₂, and MoS₂/Mo₂C (Figure R11). First, the bubble sizes of both MoS₂ and MoS₂/Mo₂C (~10⁻² mm) are about two orders of magnitude smaller than that of Pt foil (~1 mm), indicating the importance of nanostructuring on mass transfer. Second, the bubble sizes of MoS₂/Mo₂C is slightly smaller than that of MoS₂, showing surface chemistry also play a certain role on bubble

sizes. Although the differences in mass transfer between MoS₂ and MoS₂/Mo₂C is not so big, they have distinct surface chemistry and consequently, different catalytic performances.

Figure R11. Optical microscopy images showing the hydrogen bubbles on MoS₂ and MoS₂/Mo₂C surfaces. First, the bubbles sizes of both MoS₂ and MoS₂/Mo₂C ($\sim 10^{-2}$ mm) are about two orders of magnitude smaller than that of Pt foil (~ 1 mm), indicating the importance of nanostructuring on mass transfer. Second, the bubble sizes of MoS₂/Mo₂C is slightly smaller than that of MoS₂, showing surface chemistry also plays a role on bubble sizes. More details can be found in the Movie S2.

Comment 5. In the experimental part, I found that the MoS₂ microspheres were synthesized through hydrothermal reaction without high temperature sintering, while the MoS₂/Mo₂C microspheres were sintered in Ar/H₂ gas and CH₄ gas within different times. As we both known that the sintering of MoS₂ in H₂ gas can produce defects in MoS₂ with high HER activity, it is uncertain to attribute the improvement of HER activity of MoS₂/Mo₂C electrocatalyst barely to the decoration of Mo₂C.

Response 5. According to the reviewer's comment, we sintered a MoS₂ sample in Ar/H₂ gas at the same conditions for carbonization, but without introducing CH₄ (sample denoted as MoS₂-H). We tested the HER performance of the MoS₂-H sample. As shown in Figure R12, the HER performance of MoS₂-H is a little better than MoS₂ without Ar/H₂ sintering treatment but much worse than the MoS₂/Mo₂C. For example, MoS₂-H shows an overpotential of 370 mV @ 200 mA cm⁻² and MoS₂ shows an overpotential of 390 mV @ 200 mA cm⁻² in alkaline media, which is much worse than the MoS₂/Mo₂C sample (185 mV @ 200 mA cm⁻²). This control experiment suggests that Mo₂C does play an important role for the good HER performance of MoS₂/Mo₂C.

Figure R12. Polarization curves of MoS₂ samples with high-temperature Ar/H₂ sintering treatment (denoted as MoS₂-H) in KOH (1.0 M) and H₂SO₄ (0.5 M). The result shows that the HER performance of MoS₂-H is a little better than pristine MoS₂, but much worse than MoS₂/Mo₂C. This figure was added as Figure S8 in the revised Supplementary Information.

Changes to the revised manuscript. On Pages 9-10. “As a control experiment, we sintered a MoS₂ sample in Ar/H₂ at the same conditions with the carbonization experiments, but without introducing CH₄ (sample denoted as MoS₂-H). We tested the HER performance of MoS₂-H and found that though catalytic performance increases a little, it is much worse than the MoS₂/Mo₂C, indicating Mo₂C plays an important role in the good HER performance of MoS₂/Mo₂C (**Supplementary Fig. 8**).”

Comment 6. It is interesting to find that the authors assume the decorated Mo₂C nanoparticles as active sites for HER, while the ECSA of MoS₂/Mo₂C and MoS₂ microspheres are similar and the Mo₂C nanoparticles are only decorated on the terminal of MoS₂ nanosheets. Generally, people consider the active sites are linearly related with ECSA values. As MoS₂/Mo₂C and MoS₂ microspheres have similar amount of active sites, where is the improved activity come from?

Response 6. This is a good point. As we wrote in the Response 1, the catalytic performance of a catalyst is jointly determined by many factors, including intrinsic activity of each active site, numbers of active sites, mass transfer behavior, *etc.* In our experiments, although MoS₂/Mo₂C and MoS₂ have similar ECSA, the intrinsic activity of Mo₂C is higher than MoS₂, leading to a much higher overall catalytic performance of MoS₂/Mo₂C than MoS₂.

Comment 7. It is hard to understand the surface -O have higher binding energy than surface -OH in XPS, due to H⁺ has electron withdraw effect. Is it possible that the samples had been contaminated in the glovebox, as the O1s binding energy at 532.5 eV is more likely assigned as organic (-C=O-) bonds? Maybe the authors can check out with flat Pt or MoS₂/Mo₂C Pre in glove box with the same time.

Response 7. We took the reviewer’s suggestions and did the control experiment. We synthesized MoS₂/Mo₂C samples and stored in glove box for 12 hrs. Later, O 1s XPS spectrum of the sample was collected (Figure R13). We found a main peak at 530.6 eV that can be assigned to oxides, but no peak at 532.5 eV was found, suggesting that there is no organic contamination during 12 hrs storage in glove box. This result is understandable because our glove box is dedicated to store sensitive 2D materials (*e.g.*, black phosphorus), and no organic solvent is put inside.

When we compare Mo-O-H (surface -OH) with Mo-O (surface -O), the O in Mo-O-H can withdraw electrons from both Mo and H due to its large electronegativity, while O in Mo-O can only withdraw electrons from Mo. This leads to a more increase of p-orbital electron density of O in Mo-O-H than in the Mo-O. The increase of p- electron density will result in a decrease of binding energy of s-orbital electrons in O, due to the so-called screening effect. As a result, this leads to a smaller binding energy for O1s in Mo-O-H than in Mo-O case. This result is in accordance with early literature on similar materials (Geug-Tae Kim, *et al. Applied Surface Science*, **1999**, 152, 35-54).

Figure R13. O 1s XPS spectrum of a MoS₂/Mo₂C sample stored in a glovebox for 12 hrs immediately after the carbonization. The results show that there is no organic contamination during 12 hrs storage in glove box. This figure was added as Figure S18 in the revised Supplementary Information.

Comment 8. For DFT calculation, it is recommended to calculate the H absorption energy on the surface of benchmarking Pt catalyst as well, by doing so that can verify the model and methodology for your calculation.

Response 8. This is a good suggestion and we performed the calculations according to your comments. Using RPBE functional, the optimal H absorption energy of ~ -0.09 eV on Pt(111) was obtained for 0.25 ML coverage (Norskov, J. K. *et al.*, *J. Electrochem. Soc.*, **2005**, 152, J23). It has been pointed out (Bandarenka *et al. ACS Omega*, **2017**, 2, 8141–8147) that the adsorption energies of H on Pt using PBE functional are stronger than those using RPBE functional, and increasing H coverage will improve agreement between them. Accordingly, we take our results of H adsorption energy with 1ML coverage on Pt (111) using PBE functional for comparison. The H adsorption energy is ~ -0.18 eV, which is ~ 0.1 eV stronger than that using RPBE functional. These results indicate that a correction of 0.1 eV should be applied to Figure 4d after benchmarking (shown as Figure R14 below), which is included in our revised manuscript.

Figure R14. DFT calculations showing adsorption energies for H on $-OH$ terminated Mo₂C (101) as a function of adsorbed species. These results indicate that a correction of 0.1 eV should be applied to Figure 4d after benchmarking. This figure was shown as new Figure 4d in the revised manuscript.

Comment 9. In base electrolyte, the surface of Mo₂C catalyst are terminated with $-O$ groups. For HER activity in base media, H₂O dissociation is only half of the HER reaction coordination. It is suggested to calculate H absorption energy on $-O$ terminated Mo₂C as well.

Response 9. Following the reviewer's suggestion, we have calculated the H adsorption energies on $-O$ terminated Mo₂C (101) surface as shown in Figure R15. It can be seen that H adsorption energies on $-O$ terminated Mo₂C (101) surface is much stronger than on $-OH$ terminated surfaces at low-coverage cases, while it is a little weaker and closer to zero at high-coverage cases.

Figure R15. DFT calculated adsorption energies for H on $-O$ (Ox) or $-OH$ (Hyd) terminated Mo_2C (101) surface. Ox-corr and Hyd-corr stand for the adsorption energies corrected by a shift of 0.1 eV for PBE functional. This figure was added as Figure S21 in the revised Supplementary Information.

Comment 10. As the Mo_2C nanoparticles were barely decorated on the terminal of MoS_2 nanosheets and contributed negligible increase of ECSA (as to say related with active sites), would this be reasonable to consider Mo_2C as active sites for HER?

Response 10. This is a very good question. In our experiments, we found that MoS_2/Mo_2C samples have much higher catalytic performance than MoS_2 itself, Pt, and MoS_2-H , strongly showing that Mo_2C plays an important role in improving the catalytic performance. However, MoS_2 does play role as well, because its unique structure helps mass transferring and bubble release, which are important especially at large current densities. Therefore, both MoS_2 and Mo_2C components play important roles, and the catalyst is their composites.

REVIEWERS' COMMENTS:

Reviewer #1 (Remarks to the Author):

The authors have done an excellent job on addressing the comments of all 3 reviewers and I believe the work is now publishable.

Reviewer #2 (Remarks to the Author):

The manuscript has been improved after revisions. But I am still not enthusiastic about the paper since it does not provide the kind of new insight into the HER mechanism, not does it offer a guideline or demonstration of how to design a good HER catalyst.

Reviewer #3 (Remarks to the Author):

I have carefully read the authors' response letter as well as the revised manuscript and Supplementary Information. I found that the authors have well-answered all my previous comments/suggestions and revised the paper accordingly. In addition, I have read the comments and response for two other reviewers, and I believe the authors have well addressed these comments as well.

Overall, this paper developed new strategies to synthesize electrocatalysts for high performance HER. The work has solved a long-standing challenge in the HER field, i.e., high current density HER over 1000 mA/cm², which I believe will have a big impact in the field. The catalyst design and synthesis strategy reported here could excite future interests of researchers work in not only HER, but also other electrochemical reactions and systems. Therefore, I am happy to recommend a publication of the work in Nature Communications after the authors improve the follow tow minor points.

1. The dotted lines in Figure 2b are not clear to read for readers. I suggest using thicker lines. In addition, the unit in Figure 2c should be in brackets to make it consistent with other figures.

2. The names of samples in the Supplementary Movies S1 and S2 and not consistent, one reads as MoS₂/Mo₂C while another one reads as Mo₂C@MoS₂. The authors should make them the same. I think MoS₂/Mo₂C is better.

Response to Reviewer #1

Comment. The authors have done an excellent job on addressing the comments of all 3 reviewers and I believe the work is now publishable.

Response. Thank you very much for your positive recommendations.

Response to Reviewer #2

Comment. The manuscript has been improved after revisions. But I am still not enthusiastic about the paper since it does not provide the kind of new insight into the HER mechanism, not does it offer a guideline or demonstration of how to design a good HER catalyst.

Response. Thank you very much for the positive recommendations. Our work shows that both surface chemistry and morphology of electrocatalysts should be considered toward high current density HER, which is a general guideline for high performance HER and many other electrochemical processes. Regarding surface chemistry, our results indicate that interactions between electrochemical conditions and Mo_2C leads to a change in the spontaneously formed surface oxygen on Mo_2C , which promotes initial discharge of water (in alkaline) and hydronium ions (in acidic). For electrocatalysts sharing similar chemical properties with Mo_2C , *e.g.*, metal carbides, metal nitrides, and MXenes, the mechanism we found should be instructive. Besides, this mechanism is essential to understand HER in different pH conditions as well as many other electrochemical reactions. Therefore, we believe our manuscript offers new insight into the HER mechanism and provide general guideline for catalyst design.

We note that in the first round comments, the Reviewer #2 wrote that “This work is interesting and instructive.”

Response to Reviewer #3

Comment. I have carefully read the authors’ response letter as well as the revised manuscript and Supplementary Information. I found that the authors have well-answered all my previous comments/suggestions and revised the paper accordingly. In addition, I have read the comments and response for two other reviewers, and I believe the authors have well addressed these comments as well.

Overall, this paper developed new strategies to synthesize electrocatalysts for high performance HER. The work has solved a long-standing challenge in the HER field, *i.e.*, high current density HER over 1000 mA/cm^2 , which I believe will have a big impact in the field. The catalyst design and synthesis strategy reported here could excite future interests of researchers work in not only HER, but also other electrochemical reactions and systems. Therefore, I am happy to recommend a publication of the work in Nature Communications after the authors improve the follow two minor points.

Response. Thank you very much for your positive recommendations.

Comment 1. The dotted lines in Figure 2b are not clear to read for readers. I suggest using thicker lines. In addition, the unit in Figure 2c should be in brackets to make it consistent with other figures.

Response. We have modified Figure 2b and Figure 2c accordingly.

Comment 2. The names of samples in the Supplementary Movies S1 and S2 are not consistent, one reads as MoS₂/Mo₂C while another one reads as Mo₂C@MoS₂. The authors should make them the same. I think MoS₂/Mo₂C is better.

Response. We have changed the name of Mo₂C@MoS₂ in Movie S1 into MoS₂/Mo₂C to make it consistent.